# Testing Kissick’s Iron Triangle—Structural Equation Modeling Analysis of a Practical Theory

**DOI:** 10.3390/healthcare9121753

**Published:** 2021-12-18

**Authors:** Brad Beauvais, Clemens Scott Kruse, Lawrence Fulton, Matthew Brooks, Michael Mileski, Kim Lee, Zo Ramamonjiarivelo, Ramalingam Shanmugam

**Affiliations:** School of Health Administration, Texas State University, San Marcos, TX 78666, USA; scottkruse@txstate.edu (C.S.K.); larry.fulton@txstate.edu (L.F.); mbrooks@txstate.edu (M.B.); mileski@txstate.edu (M.M.); kim.lee@txstate.edu (K.L.); zhr3@txstate.edu (Z.R.); shanmugam@txstate.edu (R.S.)

**Keywords:** cost containment, quality, access

## Abstract

Background/Purpose: The purpose of this research is to determine if the tradeoffs that Kissick proposed among cost containment, quality, and access remain as rigidly interconnected as originally conceived in the contemporary health care context. Although many have relied on the Kissick model to advocate for health policy decisions, to our knowledge the model has never been empirically tested. Some have called for policy makers to come to terms with the premise of the Kissick model tradeoffs, while others have questioned the model, given the proliferation of quality-enhancing initiatives, automation, and information technology in the health care industry. One wonders whether these evolutionary changes alter or disrupt the originality of the Kissick paradigms themselves. Methods: Structural equation modeling (SEM) was used to evaluate the Kissick hypothetical relationships among the unobserved constructs of cost, quality, and access in hospitals for the year 2018. Hospital data were obtained from Definitive Healthcare, a subscription site that contains Medicare data as well as non-Medicare data for networks, hospitals, and clinics (final *n* = 2766). Results: Reporting significant net effects as defined by our chosen study variables, we find that as quality increases, costs increase, as access increases, quality increases, and as access increases, costs increase. Policy and Practice Implications: Our findings lend continued relevance to a balanced approach to health care policy reform efforts. Simultaneously bending the health care cost curve, increasing access to care, and advancing quality of care is as challenging now as it was when the Kissick model was originally conceived.

## 1. Introduction

William Kissick’s health care “Iron Triangle” has been a staple in health management literature since it was first introduced in the 1994 book *Medicine’s Dilemmas: Infinite Needs Versus Finite Resources* [1]. The framework conceptually explains the behavior of three quintessential aspects of health care: cost, quality, and access. Kissick’s theory posits, in the contemporary health care environment, these three components are essentially competing aspects of the health care delivery process. He further asserts that an advantage in one leg results in a disadvantage in at least one other leg. Thus, the framework is characterized as “iron”, because it is typically challenging—if not impossible—to simultaneously achieve a low-cost, high-quality, open access health care system.

### 1.1. Background

Kissick’s work emerged in the early 1990s during another era of health care reform as well as prolific adoption of managed care methodologies in an attempt to contain costs, improve quality, and increase access. His model predicts that adequately blending each of these factors is problematic—cost containment often results in diminished quality and decreased access to care for those who need it the most. Partly as a result of the consumer dissatisfaction with these issues, policy and legislative changes to the health care industry have slowly emerged, one of which was the Patient Protection and Affordable Care Act (PPACA). Passed in 2010, the PPACA has had profound impact on the health insurance, care delivery, and provider markets for the past decade. However, concerns remain over the cost of care, effectiveness of care delivery, increasing insurance premiums, health inequity, and more. There are now calls from both sides of the political aisle to adjust the law or abolish it completely.

With the sustained and growing interest among presidential candidates and the voting public for continued improvement in the US health care system, revisiting the model takes on increased urgency. These pressures are not new and were equally poignant and divisive when Kissick first published his model. To be clear, the health care debate has been a part of the United States political landscape since Theodore Roosevelt’s re-election campaign in 1912 and has been famously considered since then by both sides of the political aisle. Yet, throughout these administrations, decades of recorded health care industry evidence, and a health care cost now eclipsing $3.6 trillion in 2019, the Kissick theory remains relatively empirically untested [2]. The purpose of this research work is to determine if the tradeoffs that Kissick proposed are realized in the data and remain as rigidly interconnected as originally conceived. Although some have called for policy makers to come to terms with the premise of the Kissick model tradeoffs [3], others have questioned the model given the proliferation of quality enhancing initiatives, automation, and technology in the health care industry and the general belief that each will disrupt the original Kissick paradigms [4,5].

Knowing the actual interrelationships among the competing factors in the model could provide legislators, health policy makers, and health care leaders much more precise insight into the tradeoffs being made—or determine if the tradeoffs continue to exist at all. Thus, we revisited the iron triangle to empirically test whether the theory—as originally conceptualized—still holds true. While we have taught this paradigm for decades throughout various courses in health care management, economics, finance, quality management, data analytics, and more, we are not certain whether the traditional Kissick model retains its historical relevance given the changes we have witnessed in the health care industry since the 1994 debut of the theory.

### 1.2. Testing the Elements of the Iron Triangle Theory

In Kissick’s original material, the author crafts a compelling tradeoff among the core elements of cost, quality, and access. Even at that point in history, he highlights varying strategies within the health care marketplace and among governmental entities in attempting to effectively manage each of these elements. In the original work, Kissick argues that as quality increases, cost containment becomes difficult. Likewise, he suggests as access increases, quality increases, and as access increases, costs also increase. However, many would argue that how one might characterize each of the original Kissick elements has evolved over time. In the section that follows, we review each of the core elements of the Kissick model, consider how each was originally defined, and discuss how emergent technologies and advancements in the practice of medicine might help clarify how the iron triangle applies in the future of health care management and policy.

### 1.3. Cost Containment

In the original model, Kissick defines the universally used “cost” as “cost containment”. Others [6] have framed this portion of the model as, “How expensive is it to deliver health care services?” On a superficial level, one might consider the construct of “cost containment” as a straightforward construct. However, in the context of the health care industry, the cost construct is more complicated than many might originally consider. To start, the perspective of cost becomes important. Notably, who is bearing the cost? Is the cost being evaluated the total cost to the entire health care system and inclusive of all stakeholders? Alternatively, is the cost being considered from the perspective of the provider, patient, or payer? In addition, is the cost tangible and directly measured, or is it intangible and indirect, such as the case with staff overload and burnout? Health care stakeholders do not bear the same cost burden, and it could be argued that much of this cost is not always directly measurable or value added to health care outcomes.

Dr. Don Berwick, former administrator of the Centers for Medicare and Medicaid Services (CMS) and CEO of the Institute for Health Care Improvement, cites that nearly twenty percent of US health care cost is attributable to overtreatment, failures in care coordination, failures in execution of care processes, administrative complexity, pricing failures, fraud, and abuse [7]. Thus, much of the “cost containment” efforts over the years have focused more on these elements, with arguably marginal, localized, and limited success. Yet, with CMS’ increased focus on value-based care and risk-based reimbursement in the United States health care industry via programs such as those within the Value-Based Purchasing program and the Medicare Access and Chip Reauthorization Act of 2015, this may be changing. Collectively, the initiatives within these two regulatory and legislative mandates include adjustment to providers’ compensation methodologies in the Medicare program and are prompting the transfer of risk for the cost of care. Thus, these programs may have a profound effect on the overall cost, quality, and access of care delivery.

### 1.4. Quality

In his original model, Kissick never clearly defines “quality,” although throughout his text he references various viewpoints on quality as a construct in various forms of health care reform. Although there is more precision regarding the framing of the other two model dimensions, quality remains somewhat elusive. This is not altogether unexpected. As the health care industry has evolved over time, the “quality” term has become more complicated to clearly define. This is not due to lack of effort. Many have formed their own opinions on the topic. For example, the Institute of Medicine defines quality as, “the degree to which health services for individuals and populations increase the likelihood of desired health outcomes and are consistent with current professional knowledge” [8]. The Joint Commission [9] indicates that quality is, “… the degree to which (health care) processes and results meet or exceed the needs and desires of the people it serves”. The Agency for Health Research and Quality (AHRQ) [10] cites the Donabedian model as a way to “…assess and compare the quality of health care organizations” (AHRQ, n.d.). Donabedian [11] postulated health care quality can be usefully conceptualized in stages, from structures, to processes, to outcomes. As an example, common measures of structural quality measures include accreditation, staff-to-patient ratio, code compliance, electronic health record meaningful use, licensures, and board certification [12]. Process quality measures are associated with the appropriateness, efficiency, and effectiveness of both technical and interpersonal methods in the provision of care. Examples of process measures include average length of stay, percentage of people receiving preventative services, clinical adherence to established clinical practice guidelines, procedure duration, and many more. Lastly, outcome measures may be aligned with changes in health status attributed to the care received—including morbidity, mortality, infections, complications, recovery time, disability, rehospitalization, patient perceptions of care, and more [13]. As the health care industry evolves, it is safe to expect the emergence of numerous additional measures—and incentives tied to their improvement—that will likely continue to alter the health care landscape.

### 1.5. Access

In the original Kissick model, “access” is framed as the unusual dichotomy that persists in the United States. Notably, those with the means to pay enjoy access to care. Those who do not have the means to pay are much more limited in their treatment options. At the time of the publication of the original book, the author notes “America consumes 14 percent of gross domestic product in health care, yet some 40 million citizens are uninsured. A health policy that guarantees access for everyone in the population is ultimately a tax policy” [1] p. 3. Of course, the United States has evolved since Kissick penned these words. One could argue that the PPACA was instrumental in expanding insurance to more Americans. As recently as 2018, the number of uninsured is now 27.9 million nonelderly individuals [14]. However, as noted previously, in the same time frame, the total percent of gross domestic product attributed to health care has risen dramatically.

One also must question whether insurance necessarily equates to access to care. While the PPACA expanded Medicaid to over 14 million individuals, there is a debate about the impact that the expansion offered to uninsured Americans [15]. Some suggest that Medicaid expansion has improved access options to the newly insured, while others suggest Medicaid beneficiaries do not enjoy the same levels of access as their commercially insured counterparts experience [16,17,18].

Thus, as with the other segments of the Kissick model, the access segment is multidimensional. Some characterize the construct as, “How easily can patients gain access to health care services?” [6]. Others characterize it as the degree to which individuals are inhibited or facilitated in their ability to gain entry to and to receive care and services from the health care system. Factors influencing this ability include geographic, architectural, mobility, and financial considerations, among others [19]. Alternatively, access can be considered to be dependent on the wants, resources, and needs individuals bring to the care-seeking process. The ability to obtain wanted or needed services may be influenced by many factors, including travel distance, waiting time, available financial resources, and availability of a regular source of care [20].

### 1.6. Research Question and Significance of the Current Study

This study is significant because, to our knowledge, this is the first study that empirically tests the tradeoffs among cost, quality, and access as Kissick posited. Given the amount of change and disruption that has occurred in the United States health care industry since the original publication of the Kissick model, in the pages that follow we seek to empirically test the validity of Kissick’s original assertions in the contemporary context. Based on Kissick’s original theoretical work, we hypothesize the following:
**Hypothesis** **1.***health care quality and the cost of care are positively associated.*
**Hypothesis** **2.***health care quality and access to care are positively associated.*
**Hypothesis** **3.***access to care and the cost of care are positively associated.*

Based on our collective knowledge of the industry and the findings of prior authors who have tested portions of the model, we conjecture that the Kissick model remains as valid today as when it was originally developed. Our assumption is that health care leaders at the local, state, and national level will be able to draw some meaningful inferences from our findings whether the model is validated or not.

## 2. Methods

The iron triangle of health care proposes that the tradeoff among cost, quality, and access is such that improvements in one area (e.g., decreasing costs) make it incredibly difficult, if not impossible, to see improvements in the other two areas (e.g., increasing quality and access). To investigate this assumption, we proposed a model where cost, quality, and access were separate constructs composed from observed variables from available secondary data.

### 2.1. Data and Sample

The Definitive Healthcare database provided the dependent and independent variables of interest in addition to the control variables for this study. The Definitive Healthcare database is a subscription repository that provides a comprehensive collection of Medicare and non-Medicare data for networks, hospitals, and clinics including facility characteristics, utilization data, cost and charges by cost center (in total and for Medicare), Medicare settlement data, and financial statement data. We limited our study sample to the complement of hospitals reporting in the Value-Based Purchasing program. This provided a more homogenous set of hospitals and further allowed us to include several contemporary cost and quality variables. All hospital data were linked based on Medicare provider number (MPN). Data for 2018 were obtained for 2766 hospitals.

### 2.2. Analysis

This is a cross-sectional, descriptive, and non-experimental study of the three primary interrelated constructs: cost, quality, and access. Given the nature of the study constructs’ relationships, structural equation modeling (SEM) was used to evaluate the directions of the hypothetical relationships among the three unobserved constructs in hospitals for the year 2018. SEM combines confirmatory factor analysis (CFA) with path analysis and regression to generate constructs from variables (CFA) and to evaluate the strength and direction of hypothetical relationships. Covariance structures among variables are evaluated as well. JASP statistical software [21] was used to evaluate the SEM. JASP uses the R statistical software [22] lavaan package [23] and combines the specifications with network graphics. Rows and columns with greater than 20% missing values were eliminated, leaving only 3% total missing. Variable medians were then imputed to complete the dataset. The full set of study variables is shown in Table 1.

All variables were min–max scaled to exist on the range of 0 to 1, as SEM is sensitive to scaling. Estimates were bootstrapped (1000 samples) to provide reliable parameter estimates. The exogenous variables in the regression models shown below include “For Profit” status, “Rural” status, and “Teaching” status. Table 2 defines the final observed and unobserved variables selected based on model fit indices and shows the unobserved constructs of cost, quality, and access in addition to the observed variables and justifications for variable inclusion. The regression models are as follows:Quality = f (Access + Cost + Total Performance Score + Hospital Compare Score + For Profit + Rural + Teaching)Access = f (Quality + Cost + Payer Mix + Staffed Beds + Occupancy Rate + For Profit + Rural + Teaching)Cost = f (Quality + Access + Operating Expense per Bed + For Profit + Rural + Teaching)

### 2.3. The Structural Equation Model

Structural equation models are divided into two parts: a structural model and a measurement model. The structural model shows potential causal dependencies between endogenous and exogenous variables in SEM; measured variables are indicated by rectangles or squares (i.e., for profit, rural, operating expense per bed, etc., in Figure 1) and latent variables are indicated by circles (i.e., cost, quality, and access in Figure 1). Error terms (“disturbances” for latent variables) are included in the SEM diagram, represented by the triangles in the model. The error terms represent residual variances within variables not accounted for by the pathways hypothesized in the model. In a traditional SEM model, the parameters of an SEM are the variances, regression coefficients, and covariances among variables. A variance can be indicated by a two-headed arrow, both ends of which point at the same variable, or more simply by a number within the variable’s drawn box or circle. Regression coefficients are represented along single-headed arrows that indicate a hypothesized pathway between two variables. These are the weights applied to variables in linear regression equations. The strength of the weights combined with the direction provides insight into the resulting aggregate effects within the analytic model. Covariances are associated with double-headed, curved arrows between two variables or error terms and indicate no directionality [24,25]. These covariances were tuned during model building. Disturbances for each construct were used to account for residual error.

Several fit metrics are often used to evaluate SEM analysis. Generally, the more fit indices applied to an SEM, the more likely that a miss-specified model will be rejected. This suggests that a combination of at least two fit indices should be used to evaluate model fit [26]. Thus, to evaluate the efficacy of the final model in our analysis we included the comparative fit index (CFI), the Bentler–Bonett normed fit index (NFI), and the root mean squared error approximation (RMSEA). CFI values over 0.90 indicate a proper fit. For the Bentler–Bonett normed fit index, values of 0.95 and above are considered to be a proper fit. RMSEAs closer to zero represent a good fit, and a model of approximately 0.08 or less suggests a reasonable model [27,28,29].

## 3. Results

Using the results of our SEM analysis provides support to the original theoretical model.

### 3.1. Descriptive Statistics

Descriptive statistics for the variables in the model are shown in Table 3. The “average” facility reported a logarithm of operating expenses per bed of 5.178, hospital compare scores slightly over 3.0, total performance scores of 37.459, occupancy rates of 57.4%, and about 240 beds. Only 18.4% of facilities were for profit, while 21.9% and 45% were rural and teaching, respectively. Many of the variables demonstrate high skewness. The number of staffed beds is particularly variable with a range of 2641 beds and a standard deviation of 215.376 beds.

### 3.2. Correlations

Figure 1 is a correlogram for the quantitative variables. The diagonal provides the histograms, while the upper diagonal provides a scatterplot and the associated correlations. The lower diagonal depicts the bivariate plots.

From this diagram, we observe the strongest positive relationship among our study variables exists between hospital compare scores (discrete on five levels, hence the shape) and total performance scores (r = 0.452, t_2764_ = 26.641, *p* < 0.001). The strongest negative relationship is between operating expense per bed and the number of staffed beds (r = −0.577, t_2764_ = −37.178, *p* < 0.001). The observed non-linear relationships depicted in some of the scatterplots in Figure 1 (e.g., occupancy rate versus beds) provide justification for a bootstrapping methodological approach. The bootstrap approach is a variant of simulation, with the major difference that repeated samples are drawn with a replacement from the dataset at hand. It is a general procedure for statistical inference based on creating a sampling distribution for a statistic by resampling; it can provide accurate answers in cases where other methods are simply not available or where the usual approximations and parametric assumptions are invalid [30].

### 3.3. Structural Equation Model Fit (Final)

The final structural equation model produced a model with improved fit indices when compared to the null model. The comparative fit index (CFI) and Bentler–Bonett normed fit index (NFI) were both 0.97 or above. Further, the RMSEA was 0.0 with a 90% confidence interval of 0.0 to 0.0, indicating the model is an excellent fit. The final SEM model study coefficients are presented in Table 4. All beta coefficients reported are standardized.

Figure 2 displays the relationships among our study variables between the main constructs and the independent variable and the construct it assists in explaining.

#### 3.3.1. Cost

Our analysis reveals a strong and significant positive relationship between both cost/quality (β: 0.052, S.E.: 0.006, *p* < 0.001) and cost/access (β: 0.0002, S.E.: 0.001, *p* < 0.001), indicating support for both Hypothesis 1 and Hypothesis 3.

#### 3.3.2. Quality

Our analysis shows a strong and significant relationship between both quality/cost (β: 0.173, S.E.: 0.010, *p* < 0.001) and quality/access (β: 0.003, S.E.: 0.001, *p* < 0.001), indicating support for both Hypothesis 1 and Hypothesis 2.

#### 3.3.3. Access

Our analysis indicates a strong and significant relationship between both access/cost (β: 0.546, S.E.: 0.013, *p* < 0.001) (Hypothesis 3 is supported) and access/quality (β: 0.144, S.E.: 0.006, *p* < 0.001), indicating support for both Hypothesis 2 and Hypothesis 3.

### 3.4. Additional Findings

Some interesting secondary findings emerged in our analysis pertaining to the organizational characteristics of our study population of hospitals. Specifically, we found that rural hospitals (β: 0.711, S.E.: 0.005, *p* < 0.001) were more costly than their urban counterparts, while teaching hospitals were associated with higher costs than their non-teaching peers (β: 0.55, S.E.: 0.006, *p* < 0.001). For-profit organizations were associated with lower overall quality (β: −0.527, S.E.: 0.010, *p* < 0.001), but demonstrated a positive association with access to care (β: 0.445, S.E.: 0.001, *p* < 0.001). Rural hospitals (β: 0.674, S.E.: 0.002, *p* < 0.001) and teaching hospitals (β: 0.417, S.E.: 0.001, *p* = 0.003) were also associated with higher overall access to care.

## 4. Discussion

This study attempts to answer the research question of whether the tradeoffs among cost containment, quality, and access in the current health care environment are consistent with the original Kissick iron triangle model.

### 4.1. Cost

In our study, cost was measured as the ratio of operating expenses to the number of staffed beds in the facility, in addition to numerous organizational characteristics. We found that as cost increases, both quality and access significantly increase. One possible interpretation of these findings is that as health care organizations enhance spending on patient care, both access and quality improve as capacity and organizational capabilities expand. Efforts to increase access to care can be an expensive endeavor. From a practical standpoint, expanding access via increasing the number of staffed beds in a hospital is costly unto itself. The single largest hospital operating expense in the short-term acute care setting is typically “salaries, wages, and benefits”, and the nursing staff is the largest segment of the labor cost structure [31]. However, increases in nurse staffing have traditionally been associated with the provision of high-quality clinical care [32,33,34,35]. Others have found that the inefficient utilization of nursing labor through extended shifts and frequent use of overtime staffing was associated with an increase in hospital mortality rate and hospital-acquired infections—both of which come with implicit and explicit costs [35]. Further, nurses working extended hours can fatigue and consequently may be more prone to commit costly medical errors. Thus, as hospitals invest in staffing—particularly investment in nursing staff—quality can be greatly enhanced.

### 4.2. Quality

Our results indicate that as quality increases, both cost and access also increase. These findings could be interpreted to mean that as quality of care improves, it comes at additional cost from the time, effort, and attention that must be dedicated to the elimination of errors, redundancies, and waste. In an environment where a “spare no expense” approach to service delivery is frequently pursued to perform heroic life-sustaining or end-of-life interventions, there is some logical basis to why the quality-of-care delivery may be positively associated with cost. It also might be inferred that as quality improves, patients, providers, and payers seek to move care to the facility to achieve superior clinical outcomes and thus augmenting access to care in the process. Given how we have defined access in our study, as a composite of the number of operational beds, occupancy rate, and payer mix, the positive relationship between quality and access may be attributed to the improved satisfaction of patients that are able to receive definitive treatment due to increased access and facility expertise. More technically robust facilities may be associated with elevated levels of access [13,36]. This facilitates the timely treatment of acute and chronic conditions. Further, since value-based purchasing and hospital compare quality scores are increasingly publicly available on CMS websites, both payers and patients are likely to prefer being treated in facilities with high CMS scores. Hospitals that are competitive in terms of quality can easily obtain more contracts from Medicare, Medicaid, and other managed care organizations; consequently, access to care increases in those hospitals. These findings also appear to support the Kissick model.

### 4.3. Access

Our analysis shows a significant and positive relationship between access and cost. This finding also aligns with the original Kissick model. This is logical given how we have chosen to define our latent variables. Specifically, we consider it reasonable to expect a direct and positive relationship between the number of staffed beds, occupancy rate, and payer mix and the operating expense per bed in the presence of controls. In extreme cases, hospitals may be asked to expend an extraordinary level of resources to provide high-quality care when more patients use their facilities. For instance, increased hospital admissions due to COVID-19 have resulted in the extreme use of personal protective equipment in addition to critical staff and equipment resources including nurses, ventilators, and appropriate medication.

Our findings also suggest a positive association between access and quality. This effect may be explained by the fact that as the number of public-facing access points increase and number referral networks grow, there can be a positive impact on patient perceptions of quality, continuity of care, and access to other hospital resources. Another logical reason why our results reflect an association between access and quality might come as a result of providers with greater access having an increased opportunity to practice and gain experience in their specialty, which may improve health care outcomes. For example, as a surgeon performs more surgeries, the mortality rate has been known to decrease [37]. These findings support the work of prior researchers who have found that with appropriate access to care, particularly in high-volume hospitals and academic medical centers, morbidity and disease progression are both curtailed [38,39,40].

### 4.4. Limitations and Recommendations for Future Research

The relationships we observed in our analysis appear to fundamentally support the Kissick model, specifically that tradeoffs exist among the cost, quality, and access constructs that Kissick originally conceptualized in this theory. Each of the studied relationships described above are statistically significant and the effect sizes we observed are sufficiently robust to be considered meaningful based on the standardized regression coefficients in Table 4 above. However, this study has some limitations, as we note below.

A primary limitation is that we used a single year of cross-sectional data. Therefore, we could not infer causality among the constructs. However, we examined two-way relationships among the constructs, which is an approach that, to our knowledge, has never been applied to test the Kissick model. Future studies might consider examining the studied relationships over time to determine if there is strengthening or weakening of effect sizes and significance as the health care industry evolves.

Additionally, our study was limited by data availability, specifically our ability to capture more granular measures of cost, quality, and access. Despite this limitation, we found a very good model fit with our approach. However, future researchers might consider developing the construct for all three of these variables differently and may prefer different inputs. For example, the “cost” construct can be considered in numerous ways and, if possible, should examine both tangible and intangible costs, inclusive of opportunity costs. Similarly, we could consider the “quality” construct differently. We chose a composite measure in the Hospital Value-Based Purchasing total performance score and a measure of patient perceptions of quality in the hospital compare data. Although these measures capture a robust set of quality inputs across Medicare cost-reporting hospitals, future work might consider the use of more granular measures of clinical outcomes. Future analysis might also consider the human factor and examine engagement data as an input to quality and cost. Prior research has shown a positive relationship between “staff engagement”, “quality”, and “retention” with a reverse relationship to “staff replacement cost” [32,33,34,41]. We also defined quality from an organizational outcomes perspective. One might also consider the “quality of life” of patients in lieu of solely focusing on costly life-saving measures at end of life. This is an example of a different perspective on quality, and this perspective is more representative of countries who embrace quality of living over costly life-saving measures. Unfortunately, our data simply do not yet support this level of detail, but we believe we may obtain different results if these variables were considered differently.

Last, this study focused on Medicare Hospital Value-Based Purchasing (HVBP) reporting from short-term acute care hospitals. Future studies might consider this same study, examining a wider set of organizations to determine if there are observed differences among other types of hospitals in the United States or in other countries. This might include non-Medicare reporting short-term acute, long-term, children’s, psychiatric, rehabilitation, or other specialty hospital settings.

## 5. Practice Implications

In the end, our findings fully support the original Kissick model. The inherent tradeoffs in the US health care system, among the constructs of cost, quality, and access, appear to remain as complex today as they were in the days of Medicare’s inception. However, we suggest the most poignant policy implication from this study is how our industry defines “value”. While we have used current and relevant measures to assess Kissick’s primary iron triangle components of cost containment, quality, and access, concern over measurement validity remains. As a specific example, the current clinical performance measures within the Hospital Value-Based Purchasing program may not be measuring “quality” or “value” as appropriately as possible. Notably, we suggest that the clinical measures highlighted in our study simply address 30-day mortality rates from highly chronic and acute conditions. We suggest there is greater cost avoidance potential in working to prevent individuals from advancing into these acute and chronic conditions. However, as of yet, HVBP-participating organizations are not evaluated on how they assist patients in avoiding chronic disease or maintaining wellness in the local community. In our view, this carries out little to mitigate the persistent cost escalation of care delivery in the United States. In an increasingly complex and costly health care future, as an industry, we must perform a better job of measuring the outcomes we want to promote. Despite the best of intentions of past and present health policy makers, we humbly suggest we still have a long way to go.

## Figures and Tables

**Figure 1 healthcare-09-01753-f001:**
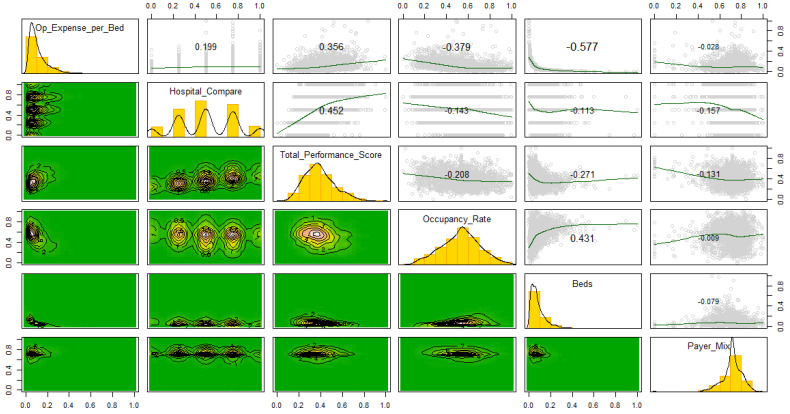
Correlogram for the quantitative variables.

**Figure 2 healthcare-09-01753-f002:**
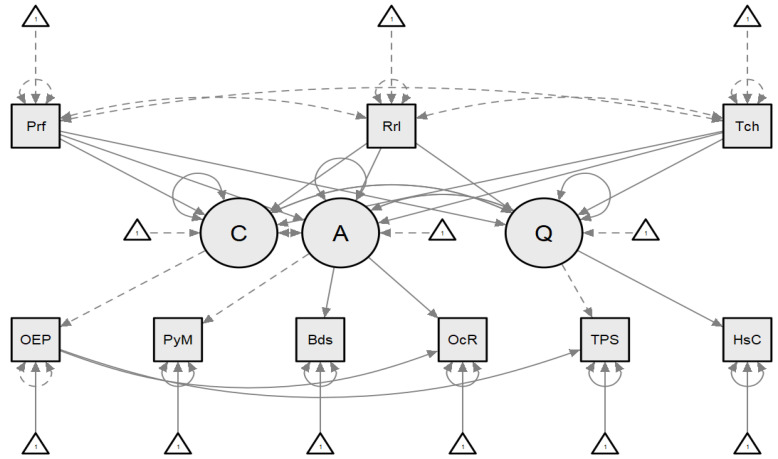
The SEM model where C = cost, A = access, Q = quality, Prf = for profit, Rrl = rural, Tch = teaching, OEP = operating expenses, PyM = payer mix, Bds = beds, OcR = occupancy Rate, TPS = total performance score, HsC = hospital compare.

**Table 1 healthcare-09-01753-t001:** Variables and operational definitions.

Variable	Original Source	Definition
Total Performance Score	Definitive Healthcare (via Medicare website)	The total performance score is the quality score used by CMS to adjust Medicare reimbursement and is an aggregate of equally weighted quality metrics from four domains in 2018: 25% safety, 25% clinical care, 25% efficiency, and 25% cost reduction, and patient and caregiver-centered experience of care/care coordination.
Hospital Compare Score	Definitive Healthcare (via Medicare website)	Hospital Compare is a consumer-oriented website owned by Medicare that provides relative scoring information on how well hospitals provide recommended care to their patients. Scored on a five-point scale.
Occupancy Rate	Definitive Healthcare	The occupancy rate is a calculation used to reflect the actual utilization of an inpatient health facility for a given time period. Occupancy rate = total number of inpatient days for a given period × 100/available beds × number of days in the period.
Payer Mix	Definitive Healthcare	Payer mix refers to the percentage of patients with government health plans—Medicare and Medicaid —vs. commercial or “private” insurance.
Natural Logarithm of Operating Expenses /Bed	Definitive Healthcare	The mean cost for each bed in the facility, a measure of cost.
Rural Status	Definitive Healthcare	A hospital located in a non-metropolitan county, or a hospital within a metropolitan county that is far away from the urban center, as defined by the Health Resource Services Administration (HRSA)
For Profit Status	Definitive Healthcare	Hospitals operated by investor-owned organizations
Teaching Status	Definitive Healthcare	Hospitals affiliated with universities, colleges, medical schools, or nursing schools.

**Table 2 healthcare-09-01753-t002:** Unobserved constructs, observed variables, and justifications.

Unobserved Construct	Observed Variables	Justification
Cost	Operating Expenses per Bed	Measure of cost that accounts for facility size in terms of beds.
Access	Number of Operational Beds	Number of beds available in the hospital equates to increased care availability.
	Occupancy Rate	Increased occupancy implies increased access to care. Alternatively, increased occupancy might indicate a lack of local market bed capacity.
	Payer Mix	Differences in payer mix equate to greater/lesser availability to care resources.
Quality	Total Performance Score	Improved performance scoring indicates higher levels of hospital performance across four quality dimensions: safety, clinical care, efficiency and cost reduction, and patient and caregiver-centered experience of care/care coordination.
	Hospital Compare	Improved scores imply elevated patient perceptions of care.

**Table 3 healthcare-09-01753-t003:** Descriptive statistics.

*n* = 2766	Mean	Median	SD	Skewness	Minimum	Maximum
ln(Op Exp./Bed)	5.178	5.160	1.039	0.076	1.596	8.597
Hospital Compare	3.054	3.000	1.114	−0.076	1.000	5.000
TPS	37.459	36.330	11.371	0.544	6.000	87.330
Occupancy Rate	0.574	0.582	0.164	−0.069	0.086	1.005
For Profit	0.184	0.000	0.388	1.629	0.000	1.000
Beds	239.893	183.000	215.376	3.174	13.000	2654.000
Rural	0.219	0.000	0.413	1.362	0.000	1.000
Payer Mix	0.709	0.714	0.112	−1.907	0.000	1.000
Teaching	0.450	0.000	0.498	0.202	0.000	1.000

Note: TPS: total performance score; ln(Op Exp./Bed): logarithm of operating expenses per bed.

**Table 4 healthcare-09-01753-t004:** SEM coefficients: statistically significant variables.

Dependent Variable	F(x)	Independent Variable	Standardized β (Lavaan)	Standard Error	*p*
Access	=~	Payer Mix	0.009		
Access	=~	Beds	−0.055	1.874	0.002
Access	=~	Occupancy Rate	−0.142	5.779	0.009
Access	~	Quality	0.144	0.006	<0.001
Access	~	Cost	0.546	0.013	<0.001
Access	~	For Profit	0.445	0.001	<0.001
Access	~	Rural	0.674	0.002	<0.001
Access	~	Teaching	0.417	0.001	0.003
Cost	~	Op Expense Per Bed	0.094		
Cost	~	Access	0.0002	0.001	<0.001
Cost	~	Quality	0.052	0.006	<0.001
Cost	~	Rural	0.711	0.005	<0.001
Cost	~	Teaching	0.550	0.006	<0.001
Quality	~	Total Perf Score	0.070		
Quality	~	Hospital Compare	0.213	0.685	<0.001
Quality	~	Access	0.003	0.001	<0.001
Quality	~	Cost	0.173	0.010	<0.001
Quality	~	Profit	−0.527	0.010	<0.001
TPS	~	Op Expense Per Bed	0.579	0.090	<0.001
Occupancy Rate	~	Op Expense Per Bed	0.431	0.006	<0.001

Note: ~ latent variable fit; =~ regression fit.

## Data Availability

Data supporting reported results for this article can be found at Definitive Healthcare at defhc.com (accessed on 19 February 2020) or via aggregation of the Centers for Medicare and Medicaid Services Cost Reports, available at: https://www.cms.gov/Research-Statistics-Data-and-Systems/Downloadable-Public-Use-Files/Cost-Reports (accessed on 19 February 2020) and the Hospital Value-Based Purchasing Program website, available at: https://www.cms.gov/Medicare/Quality-Initiatives-Patient-Assessment-Instruments/HospitalQualityInits/Hospital-Value-Based-Purchasing (accessed on 19 February 2020).

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
