# Peer review of "Testing Kissick’s Iron Triangle—Structural Equation Modeling Analysis of a Practical Theory"

_healthcare, 2021, doi:10.3390/healthcare9121753_

Round 1

Reviewer 1 Report

Manuscript can be accepted.

Reviewer 2 Report

The authors have addressed all my concerns. Their paper is ready for publication.

This manuscript is a resubmission of an earlier submission. The following is a list of the peer review reports and author responses from that submission.

Round 1

Reviewer 1 Report

Testing Kissick’s Iron Triangle - Simultaneous Equation Modeling Analysis of a Practical Theory

Brad Beauvais, Clemens Scott Kruse, Lawrence Fulton, Matthew Brooks, Michael Mileski, Kim Lee, Zo Ramamonjiarivelo and Ramalingam Shanmugam

Herewith I am submitting my reviewer comments for the above mentioned manuscript which is under consideration to be published in Healthcare.

The article is about the Kissick model (basing health care decisions on cost containment, quality, and access). The authors used Structural Equation Modeling to validate this model. Overall, I found the article quite well written and accessible. I especially liked table 1 as well since it is very useful for a non-expert. The English is of sufficient quality as far as I can tell. Overall, I find that the paper provides an interesting discussion on an important topic. It is difficult to judge the novelty for me. The model itself is very established and the study confirms it. So in a way there are not really new insights. This is probably to answer for a reviewer who is closer to the field.

Something I found a bit strange is that there is a separate pillar for cost but costs also appear in the parameters that go in the quality measures. I guess this is difficult to fix since this builds on other data but to my opinion cost is given quite a lot of weight here.

I did not understand table 3 very well. Maybe it would help to provide a bit more info in the caption. What are the units of the numbers? How can there be a mean of less than zero beds?

Line 282: “From our correlation matrix (figure not shown),” I would find this interesting to show

Author Response

We want to thank our reviewers for taking the time to critically evaluate our work. Your efforts have helped to improve our effort.

R1, Q1: The article is about the Kissick model (basing health care decisions on cost containment, quality, and access). The authors used Structural Equation Modeling to validate this model. Overall, I found the article quite well written and accessible. I especially liked table 1 as well since it is very useful for a non-expert. The English is of sufficient quality as far as I can tell. Overall, I find that the paper provides an interesting discussion on an important topic. It is difficult to judge the novelty for me. The model itself is very established and the study confirms it. So in a way there are not really new insights. This is probably to answer for a reviewer who is closer to the field.

R1, Q1, Response: We appreciate the kind words of support pertaining to our paper. We consider the effort to be ‘novel’ from the perspective that it is the first known quantitative evaluation of the Kissick cost-quality-access theoretical model.

----------------------------------------------------------------------------------------R1, Q2: Something I found a bit strange is that there is a separate pillar for cost but costs also appear in the parameters that go in the quality measures. I guess this is difficult to fix since this builds on other data but to my opinion cost is given quite a lot of weight here.

R1, Q2, Response: This is not at all strange. Per the Kissick model, all three of the primary constructs are interrelated. As costs increase, we see a corresponding increase in quality. Concurrently, with an increase in quality, the cost of care also increases.

----------------------------------------------------------------------------------------R1, Q3: I did not understand table 3 very well. Maybe it would help to provide a bit more info in the caption. What are the units of the numbers? How can there be a mean of less than zero beds?

R1, Q3, Response: Thank you for this comment.  Initially, Table 3 provided the descriptive statistics for the minimum-maximum scaled variables (e.g., transformed to be between 0 and 1 via (x-min(x)) / (max(x)-min(x).  Since SEM is not scale invariant, transformations are necessary.  However, we agree that values depicted between 0 and 1 are not very informative.  We have replaced Table 3 with descriptives of the untransformed variables and expanded our discussion accordingly.    

----------------------------------------------------------------------------------------

R1, Q4: Line 282: “From our correlation matrix (figure not shown),” I would find this interesting to show

R1, Q4, Response: Good recommendation.  We now show a correlogram with bivariate and univariate distributions along with correlations depicted.

Reviewer 2 Report

After reading this paper I have the following concerns.

This study uses data relative to one single year (2018). To what extent are results generalizable? I my perspective the Kissick’s Iron Triangle should be tested for several years. This is not only a problem of causality as authors emphasize in the conclusion section. Rather, the relations between variables may change because the context is changed. The probability that such changes occur over time is generally high.

As authors use confirmatory research (SEM methodology) they should develop hypotheses to test in addition to research questions. These hypotheses should be base on a sound literature review and logical reasoning. Hence, suggest that authors perform an in-depth review literature supporting the hypotheses that should be tested.

To what extent findings can be generalized to other countries? There are different context and institutional settings that may affect the relationships among variables.

Is there any difference between private and public hospitals?

Author Response

R2, Q1:This study uses data relative to one single year (2018).

R2, Q1, Response: This is correct and is noted as a limitation in our paper. As a preliminary study, and the first of its kind to quantitatively examine the Kissick model, we believed a single year of data was a good place to start.

----------------------------------------------------------------------------------------

R2, Q2: To what extent are results generalizable?

R2, Q2, Response: Given the scope of our study encompasses 2,766 short term acute care hospitals in (~70% of all STACs) in the United States, we consider our study is generalizable to short-term acute care hospitals in the United States.

----------------------------------------------------------------------------------------

R2, Q3: I my perspective the Kissick’s Iron Triangle should be tested for several years. This is not only a problem of causality as authors emphasize in the conclusion section. Rather, the relations between variables may change because the context is changed. The probability that such changes occur over time is generally high.

R2, Q3, Response: We agree and note this as a limitation of our current paper on lines 395 – 397 of our original draft, “Future studies might consider examining the studied relationships over time to determine if there is strengthening or weakening of effect sizes and significance as the health care industry evolves.”

----------------------------------------------------------------------------------------

R2, Q4: As authors use confirmatory research (SEM methodology) they should develop hypotheses to test in addition to research questions. These hypotheses should be base on a sound literature review and logical reasoning. Hence, suggest that authors perform an in-depth review literature supporting the hypotheses that should be tested.

R2, Q4, Response: While we understand the desire to hypothesize directionality of the relationships among the Kissick model constructs, we did not consider it to be appropriate for the current work. As the first known quantitative study of its kind to consider the matter, we refrained from injecting bias into our analysis. We allowed the data to show us (1) if a statistically significant relationship exists, and (2) the strength of the relationship, and (3) the directionality of the relationship in the presence of covariate measures. Having said this, we believe we addressed a number of salient online, peer-reviewed, and popular press sources within each of the constructs in the Kissick model to better understand our findings in the context of the broader literature. However, none of them directly address the central questions of our paper. In the end, we resolved to offer an unprecedented quantitative evaluation of the Kissick iron triangle theoretical model (Introduction, lines 28 – 37 and Background, lines 38 – 74). The logical reasoning we are testing is the theory of the Kissick model based on contemporary data.

----------------------------------------------------------------------------------------

R2, Q5: To what extent findings can be generalized to other countries? There are different context and institutional settings that may affect the relationships among variables.

R2, Q5, Response: While we consider the Kissick model theoretical tradeoffs hold true in other settings, we cannot generalize our results to other health care systems or other countries. Given our source data and the institutional settings evaluated as part of our analysis, that degree of generalizability is beyond the scope of our work. As we now note on lines 420 - 424, “Future studies might consider this same study examining a wider set of organizations to determine if there are observed differences among other types of hospitals in the United States or in other countries. This might include non-Medicare reporting short term acute, long term, children’s, psychiatric, rehabilitation, or other specialty hospital settings.”  

----------------------------------------------------------------------------------------

R2, Q6: Is there any difference between private and public hospitals?

R2, Q5, Response: We believe we partially address this question through inclusion of the ‘for profit’ variable in our study. For profit facilities are investor owned and comprise ~18.4% of our sample. All others are not-for-profit and government owned facilities. We did not separate a specific variable to exclusively examine government-owned organizations. This, too, may be a valuable focus for future research.

Round 2

Reviewer 2 Report

The authors addressed some of my concerns. However, I still invite them to consider my suggestion to develop hypotheses to test using both reasoning and literature. SEM is used to test the validity of theory using empirical data being an hypothesis-driven technique which is based on a structural model representing a set of hypotheses about the causal relations among a group of variables.